# Exploring Why Object Recognition Performance Degrades Across Income Levels and Geographies with Factor Annotations

Laura Gustafson, Megan Richards, Melissa Hall, Caner Hazirbas, Diane Bouchacourt, Mark Ibrahim

`Fundamental AI Research (FAIR), Meta`

## Abstract

Despite impressive advances in object-recognition, deep learning systems' performance degrades significantly across geographies and lower income levels—raising pressing concerns of inequity. Addressing such performance gaps remains a challenge, as little is understood about why performance degrades across incomes or geographies. We take a step in this direction by annotating images from Dollar Street, a popular benchmark of geographically and economically diverse images, labeling each image with factors such as color, shape, and background. These annotations unlock a new granular view into how objects differ across incomes/regions. We then use these object differences to pinpoint model vulnerabilities across incomes and regions. We study a range of modern vision models, finding that performance disparities are most associated with differences in *texture, occlusion*, and images with *darker lighting*. We illustrate how insights from our factor labels can surface mitigations to improve models' performance disparities. As an example, we show that mitigating a model's vulnerability to texture can improve performance on the lower income level. **We release all the factor annotations along with an interactive dashboard to facilitate research into more equitable vision systems**.

## 1 Introduction

The widespread adoption of object-recognition systems afforded by advances in deep learning comes with a responsibility: systems should work equally well across groups of individuals. Previous work demonstrates object-recognition performance is far from equal across income levels and geographies [De Vries et al., 2019, Goyal et al., 2022, Rojas et al., 2022]. This disparity encompasses publicly available recognition systems, state-of-the-art supervised and self-supervised models. Most worrisome among these findings is that the performance degradation disproportionately affects lower income households. When Artificial Intelligence (AI) systems are deployed in applications such as medical imaging, their biases can lead to disproportional harm. For example, models diagnosing COVID-19 were found to rely on geographically-biased features such as the hospital's font to diagnose patients [Roberts et al., 2021].

While existing work measures performance disparities across incomes and geographies, addressing the performance gaps remains a challenge. Key to progress is understanding *not just that, but why such disparities arise*. One hypothesis raised in DeVries et al. [2019] is that objects as well as their environments can vary drastically across regions. When factors such as object shape or lighting in a region differ from those commonly seen during training, the shift can cause model performance to drop. However, no systematic study exists characterizing how such factors vary across regions and incomes. Identifying the factors associated with model disparities can shed light on research directions to improve performance degradation across incomes and geographies.

37th Conference on Neural Information Processing Systems (NeurIPS 2023) Track on Datasets and Benchmarks.

We take a step in this direction by annotating images from Dollar Street [Rojas et al., 2022], the most common benchmark for evaluating performance disparities in object recognition systems. Dollar Street contains 38k images of household objects spanning 54 countries across income levels. We annotate each image with factors to mark what makes each distinctive, such as color, pose, shape, and texture. We first analyze how images vary across incomes and regions using our factor labels in Section 6. We find images of some classes such as *roofs* differ considerably across regions (and incomes) while others (such as *pens*) hardly vary.

We then investigate how our factor labels can explain model mistakes. We find an overall correspondence between the distribution of factors per region (and income) and model performance. Even for the latest generation of foundation models, such as CLIP [Radford et al.], performance

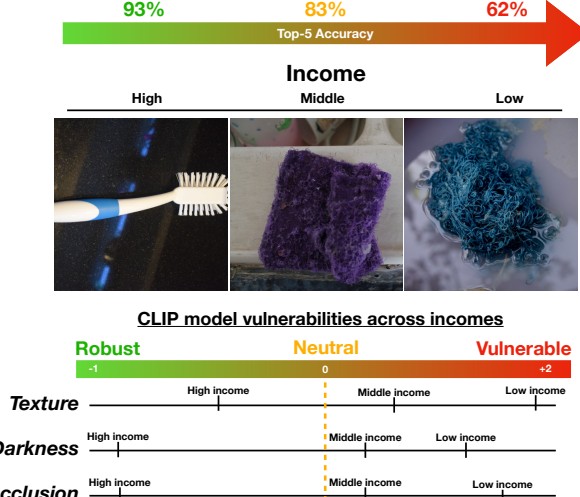

Figure 1: CLIP's vulnerability to texture, darker lighting, and occlusion are associated with performance disparities for lower incomes. We rank the most vulnerable factors based on how much more likely a factor is selected among misclassified images than overall. The example images are of *dishbrushes* from Dollar Street.

degrades by as much as 25.7% (top-5 accuracy) across incomes. We also compare the performance of other popular models across learning paradigms (self-supervised, supervised), architectures (CNN-,transformer-, MLP-based), as well as large scale pretraining [Goyal et al.]. We find remarkably similar vulnerabilities across these popular models.

Next, we precisely rank factors by examining how much more likely they are to appear among misclassifications. A factor much more likely to appear among misclassification suggests a model is vulnerable to the factor. In our analysis we find vulnerabilities in *texture*, *occlusion*, and *darker lighting* are most associated with models' performance degradation in lower incomes, Figure 1. We further study class-specific model vulnerabilities, finding strong associations between mistakes for particular classes and factors. For example, we find that for *sofas*, images labeled with texture are 7.2x more likely to appear among CLIP's mistakes than overall, suggesting texture bias is a vulnerability.

Finally, we study whether we can use robustness techniques to make fairness improvements. We show that mitigating the texture vulnerability surfaced by our analysis can improve performance disparities across incomes/regions in Section 5.2. We find a model trained to mitigate texture bias not only performs better overall +0.8% (top-5 accuracy), but on the relevant subset of images (those marked with texture), improves accuracy by 4.1% for low incomes. This suggests factor labels not only explain mistakes, but can even reveal promising mitigations to combat the disparities we observe in vision models today. Along with our analysis, we release all the factor annotations with an interactive dashboard to enable research facilitating more responsible, equitable vision system.

To summarize, our contributions are 1) we annotate all of Dollar Street images with distinctive factor labels such as pose, background, and color, 2) we explain performance disparities in models (including CLIP) using our factor annotations to reveal vulnerabilities in texture, occlusion, and darker lighting, 3) we demonstrate mitigating the vulnerability to texture can improve performance disparities across incomes and geographies, 4) we release all our factor annotations with a dashboard (Figure 2) allowing researchers to interactively query and visualize image factor labels to spur research into equitable vision systems.

## 2 Annotating Dollar Street with factor labels

The Dollar Street dataset is the most common computer vision benchmark for classifying everyday objects (e.g *armchairs, pens*) across incomes and geographies. Households across the world upload images of the specified objects. These images are labeled with the object class, location, and income of the household. The income is standardized on an international scale by DollarStreet [DSI]. We use the procedure described in [Goyal et al., 2022] to aggregate household incomes into buckets: `high`, `medium` and `low`, and group countries into regions: `Asia, Africa, Europe, The Americas`. Table 3 in Appendix A.1.1 shows the number of images per income bucket/region pair.

### 2.1 Annotation Procedure

In order to explain the degradation in model performance across incomes or geographies, annotators labeled images in Dollar Street with the factors distinguishing each image. We select all 14k images overlapping with classes in the ImageNet-21k taxonomy from Ridnik et al. [2021], using the mapping from DollarStreet classes to ImageNet synsets from Goyal et al. [2022]. We follow the same annotation procedure as in Idrissi et al. [2022]. Since it's challenging to accurately label an image in isolation, we ask annotators to label how each image differs from a fixed set of three prototypical images chosen for each class. We define prototypical images for each class as those correctly classified by a ResNet-50 model with the highest confidence. We curate a list of sixteen potential factors that can distinguish an image from the prototypical images for its class. These factors include pose, various forms of occlusion, size, style, type or breed capturing common variations in images. Specifically, this set of sixteen factors has been shown to comprehensively cover most distinctive image factors via user studies comparing the sixteen factors against free-form text responses in Idrissi et al. [2022]. A full list is shown in Figure 2. Annotators select any number of factors they believe best distinguish each image. In addition, we ask annotators to provide text descriptions to account for factors outside the sixteen factors we provide, and ask if they agree with the original class label (see 2.3 for analysis of this). A more detailed description of the annotation setup and prototypical images is in Appendix A.1.

### 2.2 Factor label statistics

We first explore how frequently each factor was selected across income levels and regions. In Figure 2, we plot the distribution of factor labels across regions and income buckets. On average, annotators chose 3.2 factors per image (standard deviation 1.2). The two most correlated factors are *color* and *pattern*, with a correlation coefficient of 0.19.

**Across all incomes, *pose*, *background*, and *pattern* are the most selected factors.** For most factors, there is only a minor difference in the frequency that a factor was selected across income buckets. For *texture*, however, there's a noteworthy difference across income levels with 10.2% of images in the `low` income bucket labeled with *texture* compared to only 4.2% for `medium` and 2.0% for `high` income buckets. This implies *texture* is 5x more likely to be selected within the `low` income bucket (relative to the `high` income bucket), a stark difference.

**The most commonly selected factors are consistent across regions and incomes.** Similar to our observation across income buckets, the most striking difference across regions is for *texture*. *Texture* is selected for 7.8% of images in `Africa` but for only 2% of images in `Europe`—a 4x difference. The similarity of emerging patterns in factors across incomes to those across regions suggests that income bucket variation differences are also exhibited across geographies. This can in part be explained by the relative rates of co-occurrence of regions and income buckets in DollarStreet, see Table 3 in Appendix A.1.1.

### 2.3 Controlling for regional differences in raters' perceptions

Challenges naturally arise when running such a large annotation procedure. In our case, there can exist regional perceptions of the semantic meaning of every object label. Indeed, the Dollar Street object class labels were originally collected from the household members who took the image, rather than assigned retroactively, which means that regional perceptions could be a source of variance in the dataset.

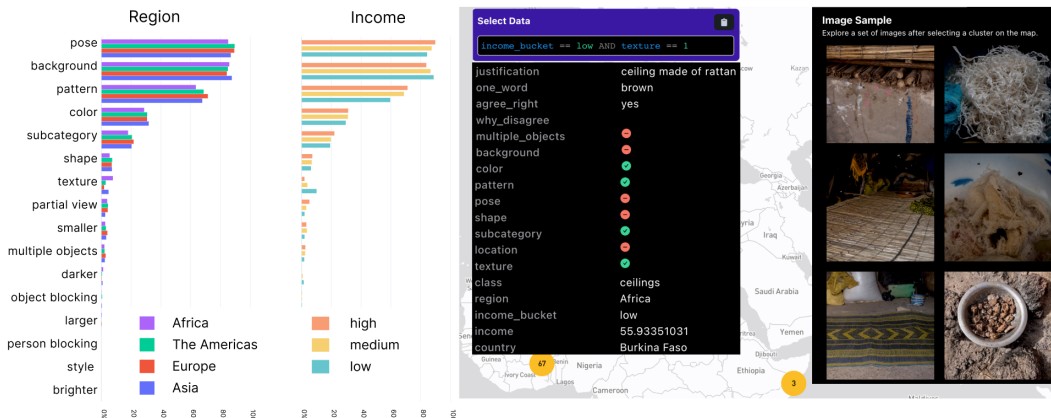

Figure 2: **Pose, background, and pattern are the most commonly selected factors**. The left panel shows the percent of images by region and income that were labelled with each factor. Annotators labelled each image with the factors that most distinguished each image from the prototypical images of its class. On the right is a screenshot of the public interactive dashboard for the annotations

For annotating DollarStreet with factor labels, we sourced 12 annotators from Indonesia. We trained the annotators on this specific annotation task before gathering the final factor label annotations. Throughout the process, we had an auditor who reviewed the annotations and brought up ambiguities. However, when gathering the factor annotations, we found that these annotators had a number of times that they disagreed with the class label. We hypothesized this could be due to differences in regional perceptions of objects.

To control for the effect of such perceptual differences across regions, we created a second annotation task. To select the images for the task, we analyzed the results of the original factor annotations, which included an option for annotators to disagree with the object label. We selected the subset of images where the annotator disagreed with the label, which totaled 2,476 images, or 18.1% of the Dollar Street dataset. For the second task, we collected additional annotations for this disputed subset, asking a different set of annotators sourced from 6 countries (5 continents) whether they agreed or disagreed with the object label. In total, there were 79 annotators who provided 12,507 label annotations. See Appendix A.1.4 for examples of annotator disagreements, and classes with the highest and lowest levels of label disagreement.

We then compared the levels of disagreement between annotators from the region where the image was taken (source region) and annotators who were from other regions. If there were region-specific biases in the label, we would expect a much higher rate of disagreement for annotators not from the source region. For images in the second task, we found annotators from both the source and other regions disagreed with the original label at similar rates (an average of 49.1% and 46.8% respectively). This consistency suggests regional differences in label perceptions do not constitute a significant source of class variation in Dollar Street. Next, we study how today's best object recognition models perform across regions and incomes.

## 3 Modern models' performance degrades across incomes and geographies

**Performance inequities are pervasive across architectures and training methods.** We compose a study on DollarStreet encompassing models across architectures (convolutional, transformer, and feedforward), learning paradigms (self-supervised, supervised, contrastive), and pretraining datasets of various sizes (up to 1 billion images). We first study the popular foundation model CLIP, which has been shown to have strong zero-shot performance on several classification benchmarks [Radford et al.]. CLIP is trained on 400M text-image pairs using a text encoder and an image encoder enabling a user to perform zero shot classification for any image. Here we prompt the model using the set of Dollar Street classes for each image to generate predictions. Our evaluation setup is described in detail in Appendix A.3. For the remaining models, to generate predictions on Dollar Street, the models are pretrained or finetuned on ImageNet21k and we use the same mapping from Goyal et al. [2022] and

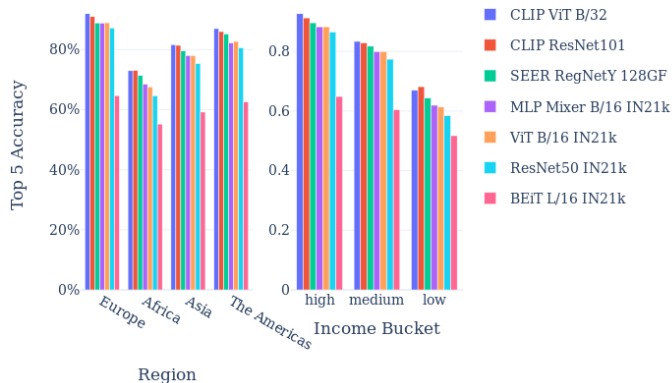

Figure 3: **Across architectures and learning procedures, model performance degrades similarly across *regions* (left) and *incomes* (right). Bars indicate top-5 accuracy.**

detail our evaluation procedure in Appendix A.3. We compare the performance of the set of models across incomes and regions in Figure 3. We find most models have comparable drops in accuracy across incomes and regions despite model differences. BEiT performs worse overall, but still exhibits similar trends in performance gaps across incomes and geographies [Bao et al., 2021]. This suggests even large scale pretraining in models such as SEER [Goyal et al., 2021], modern architectures such as ViT [Ridnik et al., 2021] or MLP Mixer [Tolstikhin et al., 2021], and self-supervised learning still don't address performance inequities. Why do such consistent disparities arise? Next we study whether factors labels can explain the performance disparities in modern vision models. We focus our study to the CLIP ViT B/32 model as it has the highest performance on DollarStreet.

## 4 Explaining model performance disparities with factor labels

We now study how our factor labels can explain the model performance disparities we observed across regions/incomes. After ruling out variables such as image quality and class imbalance in training, we demonstrate how our factor labels can surface specific model vulnerabilities associated with degradation in performance across regions/incomes.

### 4.1 Controlling for factors not captured in our annotations

**Performance disparities are not explained by image quality or training data class imbalance.** As image quality has been shown to impact the performance of facial recognition models [Xu et al., 2014], we first investigate whether image resolutions differ across regions and effect model performance. Rojas et al. [2022] found very minor differences in average image quality across region in DollarStreet. We take this a step further and find no strong correlation ($< 0.05$ Pearson's correlation coefficient) between image DPI and model performance (top-5 accuracy). Next, since class imbalance in training can skew model performance, we also investigate the extent to which class imbalance affects the disparities we observe. For ImageNet-21K pretrained or finetuned models (ViT, ResNet, MLPMixer, BEiT, and SEER), we calculate the Pearson correlation between number of images in each class for ImageNet-21K and model's top-5 accuracy. We found similarly weak correlations for all models, with coefficients less than 0.25 for top-5 accuracy (all values reported in A.4). These results suggest that variation in image quality and pretraining class imbalance explain very little of the variation model mistakes in Dollar Street. Next, we examine whether our factor labels can explain performance disparities.

### 4.2 Variation in factor labels are indicative of performance disparities

To assess whether our factor labels are indicative of model performance disparities, we measure whether larger differences in factor labels across incomes/regions correspond to larger degradations in model performance. Specifically, for each class we measure the Jensen Shannon Distances (JSD) between the factor label distributions of every pair of income buckets (and regions). This quantifies how images in a classes vary across income bucket (or region) pairs according to our factor labels.

Next, we calculate whether larger differences in images across incomes/regions correspond to larger disparities in model performance. We find as a class varies more across income pairs (according to the JSD of factor distributions), model performance gaps also increase, as shown in Figure 4. For example, classes that differ most across incomes (top quarter) suffer a 3x drop in accuracy compared to classes that differ the least across incomes (bottom quarter). We find a similar but less stark trend across regions shown in Figure 4. We acknowledge such an analysis of likelihoods is inherently not causal. Our results based on associations between factors and mistakes suggest *our factor labels are indicative of performance disparities across regions/incomes*.

### 4.3 Model performance disparities are most associated with texture, darker lighting, and occlusion

To more precisely assess which factors are most associated with mistakes, we use the same error ratio metric from Idrissi et al. [2022] to measure the association between each factor and model errors. Specifically, the error ratio for a factor quantifies how much more or less likely a factor is to appear among a model's misclassified samples as

$$\frac{P(\text{factor X} \mid \text{model errors}) - P(\text{factor X})}{P(\text{factor X})} \quad (1)$$

An error ratio greater than zero indicates how much more likely a factor is to appear among misclassified samples suggesting the factor is associated with model mistakes. For example, an error ratio of 2x indicates a factor is 2x more likely to be selected among misclassified samples than overall. An error ratio less than zero indicates a factor is less likely to appear among misclassified samples suggesting the model is robust to the factor. Since some factors are selected only for a few images, we exclude factors selected for five or fewer images in our analysis. Doing so excludes style and brightness.

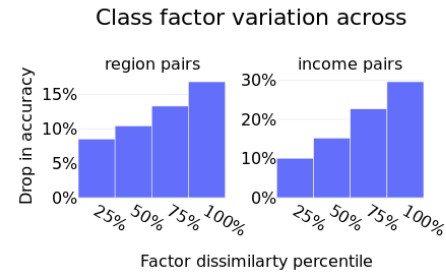

Figure 4: **Differences across regions/incomes measured by our factor labels are indicative of performance disparities.** The factor dissimilarity measures the distance (JSD) in factor label distributions across two regions/incomes for a specific class. The drop in accuracy measures the drop in performance for the given class across two regions/incomes.

**Texture, occlusion, and darker lighting are most associated with model disparities across incomes and regions.** We examine the five factors most associated with model mistakes (measured using error ratio). Overall, we find mistakes for CLIP with a ViT-B/32 encoder are most associated with *texture*, *occlusion*, or objects appearing *too small* as shown in Figure 5a. For example, *texture* appears +0.88x more among CLIP's mistakes than overall. Similarly, occlusion appears +0.76x and smaller +0.73x more so among mistakes. We conduct $\chi^2$-test to verify such differences in factor prevalence are statistically significant (see Appendix A.4).

We find these vulnerabilities also explain the performance disparities we observe for `low` incomes and regions with lower performance (`Africa`). In Figure 5a we show the five factors most associated with model mistakes across incomes (and across regions in Figure 5b). We find in the `low` income bucket, texture has the largest error ratio with *texture* 1.7x more likely to be selected among misclassifications in the `low` income bucket. On the other hand, for the `high` income bucket misclassifications are associated with quite different factors. For example, smaller objects are most associated with mistakes in the `high` income bucket. We find a similar trend for across regions with *texture*, *occlusion*, and darker lighting most associated with mistakes in the region of `Africa`. We also detail model strengths by measuring factors that are much less likely to appear among mistakes in Appendix A.4. For example, we find CLIP is much less likely to misclassify images with *partial views* of objects.

#### 4.3.1 Some model vulnerabilities are class-specific

Beyond explaining performance disparities across regions or income buckets overall, our factor labels enable us to explain vulnerabilities for specific classes. We show in Table 1 the factors most associated

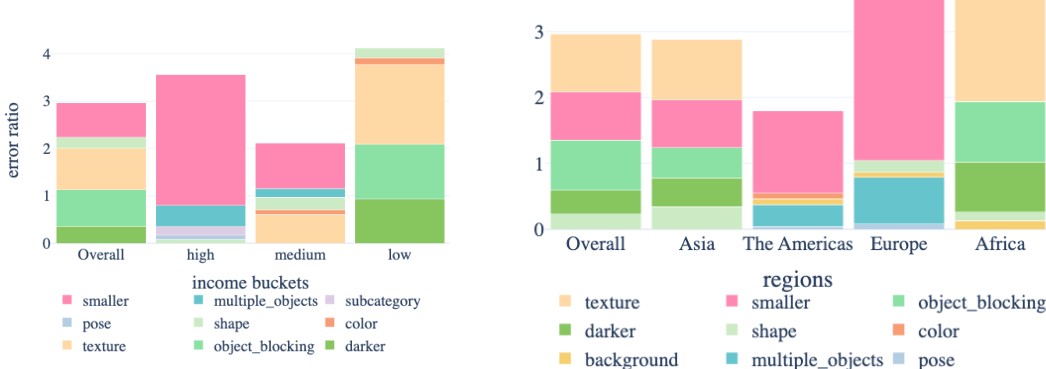

(a) Most vulnerable to **texture, occlusion, and darker lighting** for low incomes.

(b) Most vulnerable to **texture and occlusion** in Africa and Asia.

Figure 5: Figure shows the five factors most associated with CLIP's mistakes across incomes/regions. The area within each bar represents the error ratio measuring how much more likely a factor is selected among the model's misclassified samples.

| Class | Region | Factors most associated with mistakes |
|---|---|---|
| shaving | Africa | shape (+5.8x), pattern (+0.2x), background (+0.1x) |
| sofas | Africa | texture (+7.2x), pattern (+0.3x), pose (+0.1x) |
| bathrooms | Africa | background (+1.1x), pose (+0.2x), color (+0.1x) |
| kitchen sinks | Africa | color (+0.6x), background (+0.2x), pose (+0.2x) |
| showers | Africa | background (+1.2x), pose (+0.4x), multiple objects (-1.0x) |

Table 1: Class-specific vulnerabilities surfaced for the classes with lowest regional performance.

with mistakes for classes with the lowest performance across regions. We find the lowest performance is for images in the region of Africa with each class exhibiting class-specific vulnerabilities. For *shaving*, *shape* is most associated with the low performance as it's 5.8x more likely to appear among mistakes in Africa. For *sofas*, *texture* is 7.2x more likely to appear among mistakes. We find a similar pattern for classes with the low performance across incomes in Appendix A.4 and A.3. We also explore vulnerabilities at the more granular country level in Appendix A.4. These strong shifts in error ratios point to class-specific vulnerabilities.

Next we study vulnerabilities across a range of model architectures and learning procedures.

## 5 The effect of architecture and training procedure on mistake types

The factor labels also allow us to compare vulnerabilities across models. We first study vulnerabilities across a range of vision models then we show mitigating the most common vulnerability (to *texture*) can improve performance disparities across incomes and geographies.

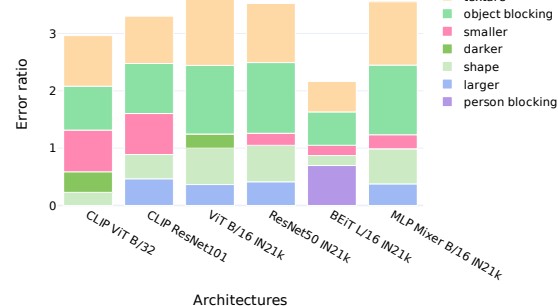

### 5.1 Comparing vulnerabilities across architectures and training procedures

In Figure 6, we examine model vulnerabilities across architectures and training methodologies. We find that, for all models, *texture*, *occlusion*, and *shape* are con-

Figure 6: **Model vulnerabilities are similar across architectures and learning procedures.** The factors shown are most associated with the model's mistakes. Bar area indicates error ratio.

sistently among the factors most associated with model mistakes. While texture is known to be a bias specifically for convolutional-models [Geirhos et al., Hermann et al., 2020], we find regardless of architecture or training procedure models have similar vulnerabilities.

### 5.2 Improving performance disparities by mitigating texture bias

Our analysis in Section 4.3 reveals *texture* is most associated with model's performance discrepancy across incomes and geographies. Can we improve this performance disparity by mitigating models' reliance on texture?

To assess this, we compare in Table 2 the performance of a standard ResNet-50 trained on ImageNet-1k compared to a ResNet-50 trained to mitigate texture bias [Geirhos et al.]. Since these are trained on ImageNet-1k (1 million images) rather than ImageNet-21k (14 million images), the overall performance is lower than other models we studied earlier (see Appendix A.5). Controlling for pre-training data, we observe a boost of +0.8% in overall top-5 accuracy for the model mitigating texture bias. On the relevant

| Top-5 Accuracy | Overall | Images Marked with Texture | | |
| --- | --- | --- | --- | --- |
| | | low income | Africa | Asia |
| ResNet-50 | 32.2 | 25.2 | 18.6 | 24.4 |
| Texture debiased | **33.0** | **29.3** | **21.6** | **25.0** |

Table 2: Texture debiasing [Geirhos et al.] can improve performance across low income buckets and regions with lower performance for images marked with texture as a distinctive factor.

subset of images (those marked with texture as a distinctive factor), we find consistent improvements in accuracy for the `low` income bucket +4.1% and in lower performing regions (`Africa` +3.0% and `Asia` +0.6%). This suggests that factor labels do not only explain model mistakes, but can also reveal potential mitigations to combat performance disparities.

## 6 How do objects vary across incomes and geographies?

Leveraging the factors gathered in our annotation process, we can quantitatively assess how classes differ across incomes and regions. For example, we can identify classes that change the most across incomes (e.g. ceiling), as well as the corresponding factors that best differentiated `high` or `low` income ceilings (pose, subcategory, and texture).

To compare how classes differ across two regions (or incomes), we compute the distribution of factors for each class. Specifically, for every pair of regions (or incomes) we normalize the distributions per class then measure the Jensen-Shannnon Distance (JSD) to characterize how images differ across regions (or incomes). Note the Jensen-Shannon Distance is the square root of Jensen-Shannon divergence between two distributions, and is a standard metric to compare discrete distributions [Endres and Schindelin, 2003]. A large JSD distance between two regions for a class indicates images differ across those regions.

**Some classes' factors vary significantly across incomes and regions; others remain consistent.** In Table 7a, we show the most starkly different classes across incomes/regions, along with their most distinguishing factors. Consistently, we find the largest differences are between `low` and `high` incomes, but don't find a consistent pattern with regions. For classes with the largest differences across regions/incomes, we find these differences to be significant with an average JSD more than twice the median JSD of all classes (across incomes/regions). Among classes differing most across regions are those relating to animals (*chickens, pet foods*); those differing most across incomes relate to building structures (*roofs, ceiling, floors*). We report full tables for the most similar and dissimilar classes across regions and incomes in Appendix A.2. In contrast, other classes don't vary across incomes/regions such as *vegetable pots*, *phones*, and *pens* as shown in Figure 7b. This suggests that while some classes can vary drastically across incomes or regions, others are quite similar.

## 7 Related work

Rojas et al. [2022], Singh et al. [2022], Goyal et al., 2022] have used Dollar Street to understand disparities in model performance between geographic and income groups, finding that many models

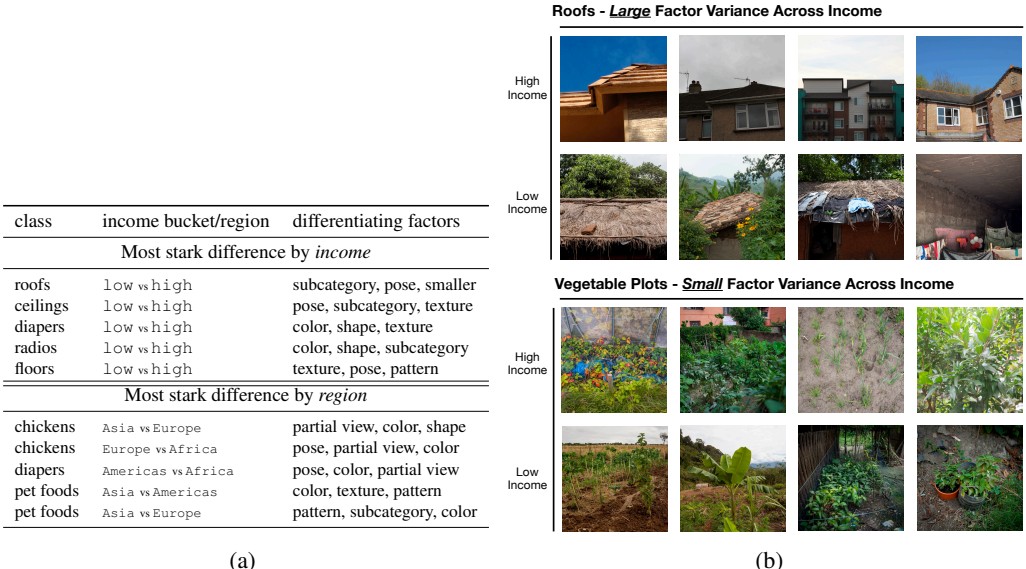

| class | income bucket/region | differentiating factors |
|-------|---------------------|------------------------|
| *Most stark difference by income* | | |
| roofs | low vs high | subcategory, pose, smaller |
| ceilings | low vs high | pose, subcategory, texture |
| diapers | low vs high | color, shape, texture |
| radios | low vs high | color, shape, subcategory |
| floors | low vs high | texture, pose, pattern |
| *Most stark difference by region* | | |
| chickens | Asia vs Europe | partial view, color, shape |
| chickens | Europe vs Africa | pose, partial view, color |
| diapers | Americas vs Africa | pose, color, partial view |
| pet foods | Asia vs Americas | color, texture, pattern |
| pet foods | Asia vs Europe | pattern, subcategory, color |

(a)        (b)

Figure 7: Classes with most stark differences in factor distributions by Jensen-Shannon Distance (JSD) across incomes/regions are listed in 7a. Examples for roofs, the class with the largest JSD across incomes, and vegetable plots, the class with the smallest JSD across income are shown in 7b.

perform better on images from Europe and the Americas, as well as those from higher household incomes. While many of the aforementioned works focus on how model architectures affect disparity findings, additional studies [De Vries et al., 2019, Shankar et al., 2017] investigate the dataset itself to identify causes of variation in model performance, including the broader geographical distribution of images as compared to the model's training data. A number of datasets [Ramaswamy et al., 2023, Dubey et al., 2021, Kim et al., 2021] and dataset auditing tools [Wang et al., 2022] have since been developed with geographic diversity in mind. De Vries et al. [2019] investigated the use of English as a "base language" for data collection. Empirical studies have shown that the "concreteness" of English words can vary greatly where crowd-sourced annotators consider words like "human" and "bobsled" more concrete than words like "recreation" and "outage" [Brysbaert et al., 2014], and previous discussions of "class label perceptions" distinguish physical properties of a substance (such as orientation or texture) from a purpose relative to the specific being that interacts with the substance, such as being "sit-on-able" [Gibson, 1979]. The relationship between the true meaning of a concept versus its perceptible form remains contested for both models [Bender and Koller, 2020] and humans [Phillips, 2019]. Beyond Dollar Street, other works study variations in representations of concepts across visual factors including pose, background, occlusion, etc. within ImageNet [Idrissi et al., 2022] and collect supplementary multi-class labels [Yun et al., 2021, Beyer et al., 2020, Shankar et al., 2020]. Barbu et al. [2019] created a benchmark to measure a model's robustness to backgrounds, rotations, and viewpoints.

## 8 Conclusion

In this work, we take a step towards explaining why disparities in object-recognition systems arise. We annotate images from the Dollar Street dataset with distinguishing factors in order to explain how objects differ across incomes and geographies. Using these labels, we identify vulnerabilities in CLIP, a foundation model with impressive zero-shot classification performance. We find disparities in model performance are associated with texture, occlusion, and darker lighting. Finally, we surface initial promising mitigations such as texture debiasing that can improve performance disparities. This shines light on a promising research direction leveraging techniques in robustness for fairness gains. In future work, we plan to explore further targeted mitigations can improve performance disparities in vision systems. While our conclusions are limited by the number of samples and representative diversity of the Dollar Street dataset, we hope by releasing our factor annotations we spur further research into equitable vision systems.

## 8.1 Limitations

While grounding our new annotations in the widely used Dollar Street dataset allows for easy adopting and extensibility of existing analyses, we are also exposed to limitations of the underlying dataset. For example, Dollar Street contains imbalances across object classes and some subjectivity in class definitions across groups (e.g. *most loved items*, *nicest shoes*). We attempt to address these gaps by filtering out classes with few samples and using only classes with an ImageNet mapping.

In addition, annotators from different regions may have different perceptions of what constitutes each image factor and object class. We address this by providing annotators with examples of image factors and conducting an additional round of annotation focused on understanding regional variations in rater perceptions (see Section 2.3 and Appendix A.1.4). However, there may be lingering discrepancies left undiscovered.

Furthermore, the factors we annotate (e.g. color, shape, pose) are only a subset of factors that can relate to model performance. While these factors are relevant to standard computer vision benchmarks [Idrissi et al., 2022], additional factors may prove insightful, especially for domain specific-tasks. For example, annotations of foliage and shadows may be useful for auditing autonomous driving models.

Finally, our analyses are only a first attempt at explaining discrepancies and performing mitigations. We hope that our rich annotations and dashboard for dataset understanding enable future studies into evaluations and mitigations of geographic disparities.

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

# A  Appendix

## A.1  Annotating Dollar Street with factor labels

### A.1.1  DollarStreet Statistics

| Region | Income Level | | |
|---|---|---|---|
| | low | medium | high |
| Africa | 2141 | 1443 | 280 |
| Asia | 1362 | 8673 | 1424 |
| Europe | 0 | 1443 | 1455 |
| The Americas | 339 | 2093 | 1223 |

Table 3: Number of images for each region, income level pair in Dollar Street.

Table 3 shows the number of images in Dollar Street for each income and region pairing. We observe the distribution across images and regions is far from uniform, implying region and income distributions skew of counts are entangled. Consequently, we present both region and income comparisons where appropriate in our analysis.

### A.1.2  Prototypical Image Selection

We define prototypical images for each class as those correctly classified by ResNet-50 model with the highest confidence. We use a ResNet-50 model pre-trained on ImageNet21k from Ridnik et al. [2021]. We select the ImageNet classes that overlap with Dollar Street labels, using the mapping as defined in [Goyal et al., 2022]. We use a soft-max over the sub-section of ImageNet classes that are in the mapping. We take the top predictions and confidence for these ImageNet classes and use the defined mapping from IN21k to Dollar Street in order to make DollarStreet class predictions. Out of the box, the model does not perform well on DollarStreet. Running a full pass over the dataset with Batch Norm in train mode, without any updates to the model weights, helps with the distribution shift from ImageNet to DollarStreet images, meaning overall accuracy is higher.

We select the three images that the model predicts successfully with the highest confidence. If such images do not exist, prototypical images are hand-selected. Table 4 shows the prototypical images used for five classes.

### A.1.3  Annotation Setup

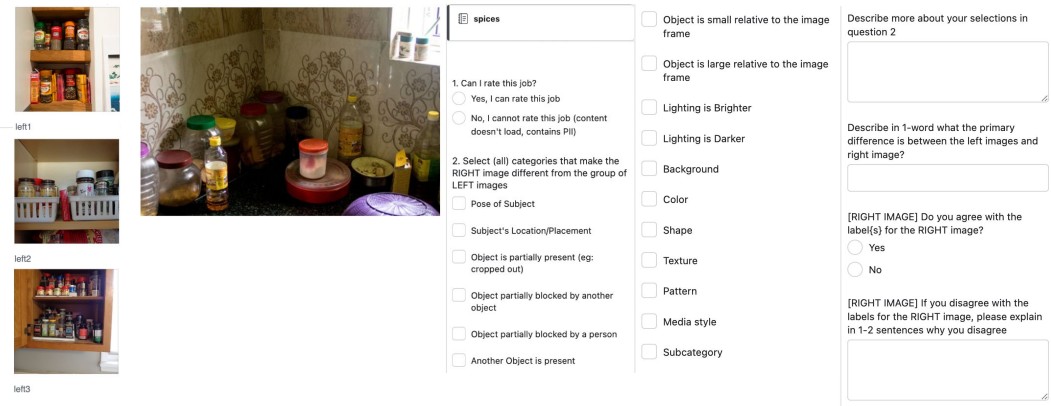

Figure 8: Example annotation task.

| Class | Prototypical Images | | |
|---|---|---|---|
| grains |  |  |  |
| plates |  |  |  |
| power outlets |  |  |  |
| cleaning floors |  |  |  |
| toothbrushes |  |  |  |

Table 4: Prototypical images used for five classes.

Figure 8 shows an example of the annotation task. Annotators select the factors distinguishing each image among sixteen factors such as pose, various forms of occlusion, size, style, type or breed. Annotators can select any number of distinctive factors for each image. We source 10 annotators through a third party vendor from South East Asia. In addition, we ask annotators to provide text descriptions to account for factors outside the sixteen we provide. We trained annotators with examples so that they were familiar with the task before annotating the target images. We had intermediate QA from the third party vendor monitoring annotations for quality. We also ask annotators whether they agree with the original class label for each image.

### A.1.4   Label Agreement Annotation Setup

| Country | Number of annotators |
|---|---|
| India | 8 |
| Nigeria | 9 |
| Brazil | 13 |
| United Arab Emirates | 6 |
| United States | 8 |

Table 5: Annotator breakdown for label agreement task.

For our follow up annotations about label agreement, we sourced 44 annotators from 5 different countries, with the full demographics shown in Table 5. We asked one annotator per country about each image in question. In Table 9, we show example images from the three most disputed classes, along with alternative labels suggested by annotators. In Table 6, we show the classes with the highest and lowest levels of disagreement among annotators.

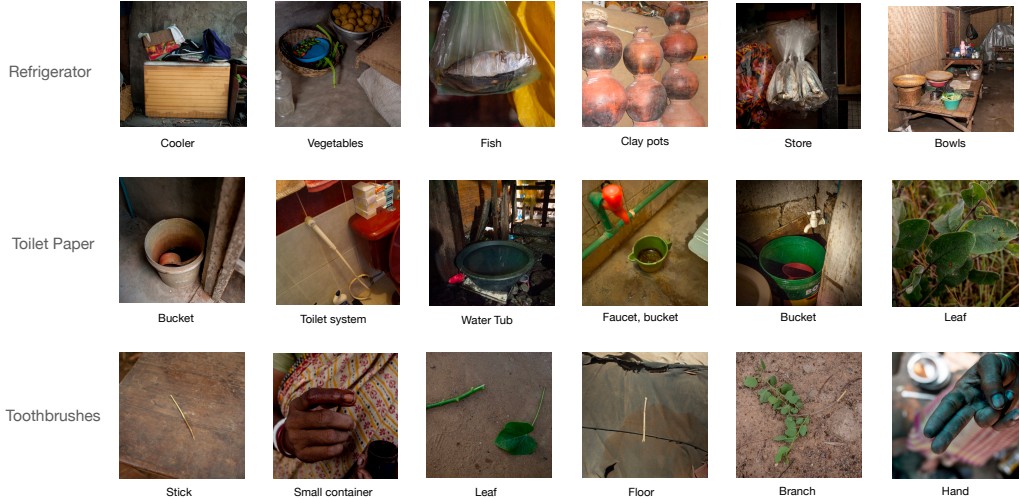

Figure 9: Randomly sampled example images and alternative labels given for the three classes with most disagreement. The original class label is shown on the left, and the alternative label given by the annotator shown below each image.

| class | % disagreement | class | % disagreement |
|---|---|---|---|
| toilet paper | 88.4 | medication | 10.0 |
| refrigerators | 83.5 | fruit trees | 11.8 |
| toothbrushes | 79.3 | plates of food | 12.3 |
| sofas | 77.8 | trash | 14.3 |
| diapers | 72.0 | cleaning floors | 14.5 |
| armchairs | 70.6 | ceilings | 15.0 |
| showers | 66.3 | homes | 15.0 |
| kitchen sinks | 64.5 | books | 15.0 |
| wall clocks | 63.2 | cooking pots | 19.8 |
| radios | 60.4 | wheel barrows | 20.0 |

Table 6: Top ten classes with the highest percentage of annotators who *disagreed* (left) and *agreed* (right) with the original class label

## A.2 How do objects vary across incomes and geographies?

We show the most dissimilar classes across incomes and regions by comparing the Jensen-Shannon Distance of the factor annotation distributions in Tables 7 and 8.

| class | income bucket | differentiating factors |
|-------|---------------|-------------------------|
| roofs | low vs. high | subcategory, pose, smaller |
| ceilings | low vs. high | pose, subcategory, texture |
| diapers | low vs. high | color, shape, texture |
| radios | low vs. high | color, shape, subcategory |
| floors | low vs. high | texture, pose, pattern |
| sofas | low vs. high | color, texture, multiple objects |
| kitchen sinks | low vs. high | shape, pose, background |
| toilet paper | low vs. high | color, pose, background |
| wardrobes | low vs. high | background, pattern, color |
| mosquito protections | low vs. high | color, subcategory, pattern |

Table 7: Classes with most stark differences in factor distributions by Jensen-Shannon Distance (JSD) across incomes.

| class | regions | distinctive factors |
|-------|---------|---------------------|
| chickens | Asia vs. Europe | partial view, color, shape |
| chickens | Europe vs. Africa | pose, partial view, color |
| diapers | The Americas vs. Africa | pose, color, partial view |
| pet foods | Asia vs. The Americas | color, texture, pattern |
| pet foods | Asia vs. Europe | pattern, subcategory, color |
| ceilings | Europe vs. Africa | pose, subcategory, texture |
| roofs | Europe vs. Africa | subcategory, pose, texture |
| car keys | Asia vs. Europe | pattern, partial view, subcategory |
| make up | Europe vs. Africa | background, subcategory, pattern |
| goats | Asia vs. Africa | pattern, color, subcategory |

Table 8: Classes with most stark differences in factor distributions by Jensen-Shannon Distance (JSD) across regions

We show the most similar classes across incomes and regions using the same procedure of comparing Jensen-Shannon Distance of the factor annotation distributions in Tables 9 and 10.

| class | income buckets | distinctive factors |
|-------|----------------|---------------------|
| vegetable plots | low vs. high | multiple objects, background, color |
| phones | medium vs. high | background, pose, multiple objects |
| pens | medium vs. high | color, background, pattern |
| bikes | low vs. high | background, subcategory, smaller |
| armchairs | medium vs. high | color, background, pose |
| latest furniture bought | medium vs. high | subcategory, background, color |
| child rooms | medium vs. high | pose, pattern, color |
| wall clocks | medium vs. high | color, pose, shape |
| cooking utensils | medium vs. high | pose, shape, pattern |

Table 9: Classes most similar in factor distributions by Jensen Shannon Distance across incomes

| class | regions | distinctive factors |
|---|---|---|
| vegetable plots | The Americas vs. Africa | pose, background, pattern |
| phones | Asia vs. Europe | pose, background, color |
| pens | Europe vs. Africa | pose, color, pattern |
| wheel barrows | Europe vs. The Americas | color, pose, background |
| ceilings | Asia vs. Africa | subcategory, pattern, texture |
| pets | Asia vs. Europe | background, pattern, subcategory |
| stoves | Asia vs. Africa | subcategory, color, pattern |
| menstruation pads | Asia vs. The Americas | pose, subcategory, pattern |
| tvs | Europe vs. The Americas | partial view, subcategory, background |
| everyday shoes | Europe vs. The Americas | color, partial view, shape |

Table 10: Classes most similar in factor distributions by Jensen Shannon Distance (JSD) across regions

### A.3 Evaluation Setup

**CLIP Prompt Engineering** We use CLIP in a zero shot setting, where we prompt the model using the set of Dollar Street classes (e.g. *medication, plates of food*) for each image to generate predictions. We generate the text prompts for CLIP by combining the 80 prompt templates used in the original CLIP paper with each Dollar Street class name, substituting _ for spaces. We consider an image correctly predicted if the top 5 classes predicted by CLIP is associated with the photo. *Note: Most photos in DollarStreet have only one label, but a small subset of (638) images containing multiple class labels (e.g. (cups, plates, dish racks) and (child rooms, kids bed, beds)).*

**ImageNet21k as a shared taxonomy** For models outside of CLIP, we use ImageNet21k to ground our models in a shared taxonomy. Following Goyal et al. [2022], we map the ImageNet21k labels to DollarStreet classes. We consider the image correctly classified if any of the top 5 ImageNet21k classes predicted by the model are mapped to any of the DollarStreet classes associated with the photo. We note that the mapping is not 1:1, and multiple classes in DollarStreet have multiple classes in ImageNet 21k that map to the single class. All of the models used for evaluation excluding CLIP and SEER are trained on ImageNet 21k. SEER is pre-trained in a self-supervised manner, and the model is fine-tuned on the 108 classes in ImageNet 21k that overlap with DollarStreet prior to evaluation. For ImageNet-21k pretraining, we use models from Ridnik et al. [2021].

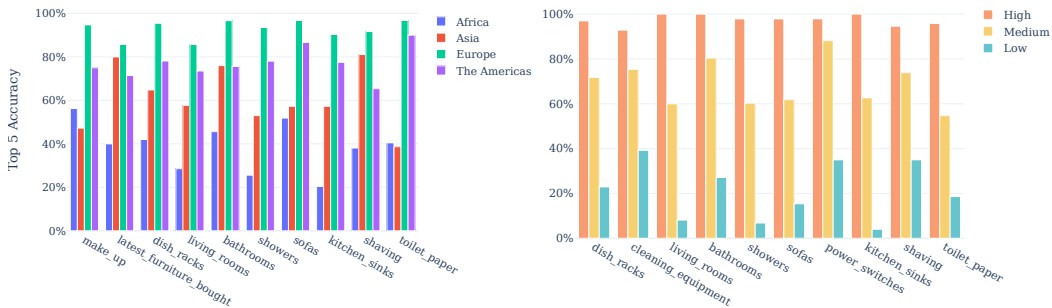

Figure 10: 10 classes with biggest performance discrepancy over regions (left) and income bucket (right).

**Class level performance disparities** Figure 10 shows the top 10 classes with the biggest performance disparity between groups for regions and incomes. We define the largest performance discrepancy as the maximum difference in accuracy between any two regions (or income buckets). At a class level, we find that the discrepancy in accuracy can be stark - over 50% for the classes with the widest gap. For both incomes and geographies, we find that the differences mostly pertain to

items in kitchens (*dish racks, kitchen sinks*) and items in bathrooms (*showers, shaving, toilet paper, bathrooms*).

## A.4 Explaining model performance disparities with factor labels

As part of our analysis of model performance disparities, we investigate the impact of pretraining class balance and image quality. In Table 11, we show the Pearson correlation coefficients and p-values between each model's top-5 accuracy and the Image DPI, a measure of image resolution. In Table 12, we show the Pearson correlation coefficients and p-values between each model's top-5 accuracy and the ImageNet-21K class count. We excluded CLIP from this analysis as CLIP was trained on a proprietary dataset.

| Model | Correlation, Top 5 Accuracy and Image DPI |
|---|---|
| ViT | -0.019 (p = 0.035) |
| ResNet50 | -0.023 (p = 0.008) |
| MLPMixer | -0.026 (p = 0.002) |
| BeIT | -0.003 (p = 0.72) |
| SEER | -0.016 (p = 0.057) |
| CLIP | -0.035 (p = 0.00005) |

Table 11: Pearson Correlation coefficients and p-values between each model's top-5 accuracy and image quality, as measured by DPI.

| Model | Correlation, Top-5 Accuracy and Class Count |
|---|---|
| ViT | 0.126 ($p < 0.0001$) |
| ResNet50 | 0.142 ($p < 0.0001$) |
| MLPMixer | 0.135 ($p < 0.0001$) |
| BeIT | 0.222 ($p < 0.0001$) |
| SEER | 0.103 ($p < 0.0001$) |

Table 12: Pearson Correlation coefficients and p-values between each model's top-5 accuracy and ImageNet-21K class counts. CLIP is not included, as it was trained on a proprietary dataset.

Factors most associated with misclassifications differ considerably across regions and incomes. We find for the `high` income bucket, objects marked as *smaller* are most associated with mistakes, appearing +2.8x more among mistakes. On the other hand, *texture* which is not among the top five factors among mistakes in the `high` income bucket is associated with mistakes in the `medium` and `low` income buckets. *Texture* is +0.6x and +1.7x more likely to appear among mistakes in the `medium` and `low` income buckets respectively. We also find in the `low` income bucket, factors such as *occlusion* and *darker lighting* to be associated with model mistakes, appearing +1.2x and +0.9x more so among mistakes in the `low` income bucket. This suggests specific factors such as *texture*, *occlusion*, and *darker lighting* are associated with the disparity in performance we observe across incomes.

**Further discussion of actors associated with mistakes across regions.** We also measured the factors associated with model mistakes across regions in Figure 5 in Section 4.3 of the main text. In `Asia` we observe the factors most associated with mistakes are similar to those associated with mistakes overall. However, we find distinctive factors are associated with mistakes across each of the other regions. In the Americas, we find *smaller objects* (+1.2x more likely to appear among mistakes), followed by images with *multiple objects* (+0.3x). Similarly in `Europe`, *smaller objects* and *multiple objects* are most associated with mistakes appearing +2.8x and +0.7x more so among mistakes respectively. In `Africa` however, we find instead *texture* (+1.6x) most associated with mistakes, followed by *occlusion* (+0.9x) and *darker lighting* (+0.8x). This suggests the disparity

due to lower performance in regions such as `Africa` are associated with distinct factors related to *texture*, *occlusion*, and *darker lighting*.

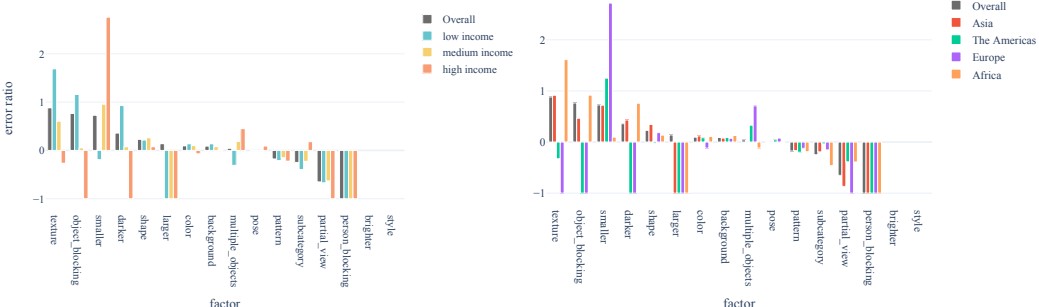

Figure 11: Shows the full error ratios for each factor per income bucket (left) and region (right). An error ratio higher than zero indicates the factor is more associated with model mistakes; less than zero indicates the factor is less likely to appear among a model's mistakes.

**Statistical significance of error ratios for top factors.** To confirm the top factors associated with model mistakes measured by our error ratio are statistically significant. We conduct a Chi-Squared test comparing the overall distribution of counts of the top factors to their distribution of counts among misclassifications. We find a statistically significant difference with a Chi-Squared statistic of 21.7 (p-value =0.0002).

| Class | Income | Factors associated with mistakes |
|---|---|---|
| sofas | low | pattern (+0.5x), background (+0.3x), pose (+0.2x) |
| toilet paper | low | texture (+3.3x), shape (+2.7x), color (+0.8x) |
| living rooms | low | background (+0.8x), pose (+0.0x), color (-0.1x) |
| kitchen sinks | low | color (+0.5x), background (+0.3x), pose (+0.2x) |
| showers | low | background (+0.9x), pose (+0.3x), pattern (-0.5x) |

Table 13: Class-specific vulnerabilities surfaced by our factor labels. We show vulnerabilities for the classes with lowest income performance.The values in parenthesis indicate how much more likely a factor is to appear for misclassified samples.

**Factors most associated with largest discrepancies for classes across income buckets.** We show the three factors most associated with model mistakes for the classes across income buckets with largest performance gap in Table 13. Trends are similar to those shown in the main paper for the largest disparity per region.

**Additional analysis of vulnerabilities by country** In Table 14 we show the most vulnerable factor by country along with its error ratio for CLIP with a ViT encoder.

**Additional analysis on the effect of architecture and training procedure** We extend our evaluation of CLIP models to include a ResNet50 encoder, to enable more consistency between the architectures of our CLIP and supervised models. Results per income are given in 15.

## A.5  Texture debiasing experimental details

To measure the effect of reducing texture bias from Geirhos et al., we create a mapping from Dollar Street classes to ImageNet-1k similar to Rojas et al. [2022]. We initialize the mapping by matching the embedding similarity of each class name to its nearest neighbors from ImageNet-1k using a pre-trained Spacy language model eng-large https://spacy.io/usage/linguistic-features#vectors-similarity. We then manually correct any

| Country | Most vulnerable factor | Error Ratio |
|---|---|---|
| Bangladesh | shape | 1.77 |
| Bolivia | color | 0.48 |
| Brazil | multiple_objects | 3.25 |
| Burkina Faso | texture | 1.56 |
| Burundi | color | 0.42 |
| Cambodia | texture | 0.7 |
| Cameroon | background | 0.23 |
| China | smaller | 3.52 |
| Colombia | color | 0.34 |
| Egypt | pattern | 0.47 |
| France | pose | 0.88 |
| Haiti | shape | 0.26 |
| India | texture | 1.2 |
| Indonesia | smaller | 2.15 |
| Cote d'Ivoire | texture | 2.17 |
| Jordan | background | 0.29 |
| Kazakhstan | background | 0.82 |
| Kenya | background | 0.51 |
| South Korea | shape | 1.71 |
| Latvia | smaller | 7.88 |
| Lebanon | background | 0.82 |
| Liberia | color | 0.49 |
| Malawi | texture | 2.27 |
| South Africa | color | 0.73 |
| Mexico | pose | 0.18 |
| Myanmar | texture | 1.84 |
| Nepal | texture | 1.65 |
| Netherlands | pose | 0.94 |
| Nigeria | texture | 1.29 |
| Pakistan | background | 1.05 |
| Palestine | background | 0.49 |
| Papua New Guinea | background | 0.29 |
| Peru | pose | 0.91 |
| Philippines | texture | 1.98 |
| Romania | background | 0.43 |
| Russia | pose | 0.42 |
| Rwanda | texture | 3.72 |
| Somalia | background | 0.58 |
| Sri Lanka | background | 2.65 |
| Sweden | pose | 0.28 |
| Thailand | subcategory | 0.16 |
| Tunisia | background | 0.4 |
| United States | smaller | 3.49 |
| Ukraine | pose | 0.38 |
| United Kingdom | pose | 0.39 |
| Vietnam | background | 0.3 |
| Zimbabwe | background | 0.55 |

Table 14: The most vulnerable factor for a CLIP ViT per country.

issues in this mapping to produce ImageNet-1k mappings for approximately half of the Dollar Street classes. Note for all other analysis we use the ImageNet-21k mapping from Goyal et al. [2022].

| Model | high | middle | low |
|---|---|---|---|
| BEiTPretrained21k | 64.9 | 60.4 | 51.7 |
| MLPMixerPretrained21k | 88.3 | 79.9 | 61.9 |
| SeerPretrained | 89.6 | 81.7 | 64.3 |
| ViTPretrained21k | 88.3 | 79.9 | 61.3 |
| ResNet50Pretrained21k | 86.5 | 77.4 | 58.4 |
| CLIP ViTB/32 | 92.6 | 83.4 | 66.9 |
| CLIP ResNet101 | 91.2 | 82.9 | 68.2 |
| CLIP ResNet50 | 95.7 | 88.8 | 73.4 |

Table 15: Comparison of model architectures across incomes. Overall, we find that while architecture and learning objective are important factors for fairness considerations, there are consistent and similar vulnerabilities across models.

# B  Data Card

We provide a data card for our annotations, following the guidance of Pushkarna et al. [2022].

---

## DollarStreet Factor Annotations

We provide annotations for Dollar Street images with distinctive factor labels such as pose, background, and color to explain performance disparities in models.

Data is available at:
https://github.com/facebookresearch/dollarstreet_factors
Data visualizer is available at:
https://dollarstreetfactors.metademolab.com/

| Overview | |
|---|---|
| Publisher | Meta |
| Authors | Laura Gustafson, Megan Richards, Melissa Hall, Caner Hazirbas, Diane Bouchacourt, Mark Ibrahim |
| Contact | dollarstreet-factors@meta.com |
| Funding & Funding Type | Fundamental AI Research |
| License | CC BY-NC 4.0 |
| **Applications** | |
| Dataset Purpose | Evaluate computer vision models robustness to common factors to help pinpoint where geographical and economical performance discrepancies arise. |
| Key Application | *Computer Vision, Robustness, Fairness* |
| Primary Motivations | We can use the factors to identify model vulnerabilities that contribute to these discrepancies. Pinpointing the vulnerabilities will help guide research into developing fairer models. |
| Intended Audience | Vision researchers aiming to analyse their trained vision models. |
| Suitable Use Case | Evaluation of Computer Vision models and analysis as to a model's strengths and weaknesses. |
| **Data Type** | |
| Primary Data Type | Annotations for existing DollarStreet dataset of images |
| Primary Annotation Type | Annotations are manually gathered from expert annotators. Annotations are booleans for each of the factors, along with single word and paragraph responses detailing the annotator's logic. |
| Data SnapShot | Dataset contains
• Annotations for **14k** images
• Each image is annotated with **16** factors |
| Data Sources | Annotations were manually gathered. Annotations are for images from the existing public DollarStreet dataset. https://www.gapminder.org/dollar-street Images are licensed under CC-BY. |

| DollarStreet Factor Annotations | |
|---|---|
| Annotation format | Each item in the annotation file will contain: |

Each item in the annotation file will contain:

1. Image information:

   - `url`: Public url of image
   - `full_image_id`: Unique ID of image
   - `household_id`: Unique ID of household who took the photo
   - `class`: Image classification class

2. Group information:

   - `region`: Region where the image is from. Options are *The Americas, Europe, Africa, Asia*. Derived from `country`.
   - `income_bucket`: Income bucket of household who took the image. Options are *high income, middle income, low income*. Derived from `income`
   - `country`: String of country name where image is taken.
   - `lat`: Latitude of country
   - `lng`: Longitude of country
   - `income`: Integer of income of household. TODO metric

3. Summary:

   - `one_word`: One word describing how the image differs from the prototypical images for it's class.
   - `justification`: String description for the annotators' justification of their one word summary.
   - `agree_right`: Boolean describing whether the annotator agreed with the class label
   - `why_disagree`: If `agree_right` is `False` this will contain a string explanation as to why the annotator disagreed with the class label

4. Factors (Boolean):

   - `multiple_objects`
   - `background`
   - `color`
   - `brighter`
   - `darker`
   - `style`
   - `larger`
   - `smaller`
   - `object_blocking`
   - `person_blocking`
   - `partial_view`
   - `pattern`
   - `pose`
   - `shape`
   - `subcategory`
   - `location`
   - `texture`

## B.1 Interactive factor dashboard

We show screenshots of our interactive dashboard for exploring the factor labels across regions in Figures 12 and 13. The dashboard allows for interactive queries by region, income, factor label. Each query yields sample images, which you can interactively explore annotations for as shown in 13. We hope this tool will allow researchers to easily explore factor labels associated with images across axes such as regions or incomes to spur further research into reliable vision systems.

Figure 12: Interactive dashboard for Dollar Street factor annotations with an income and factor label query (for texture).

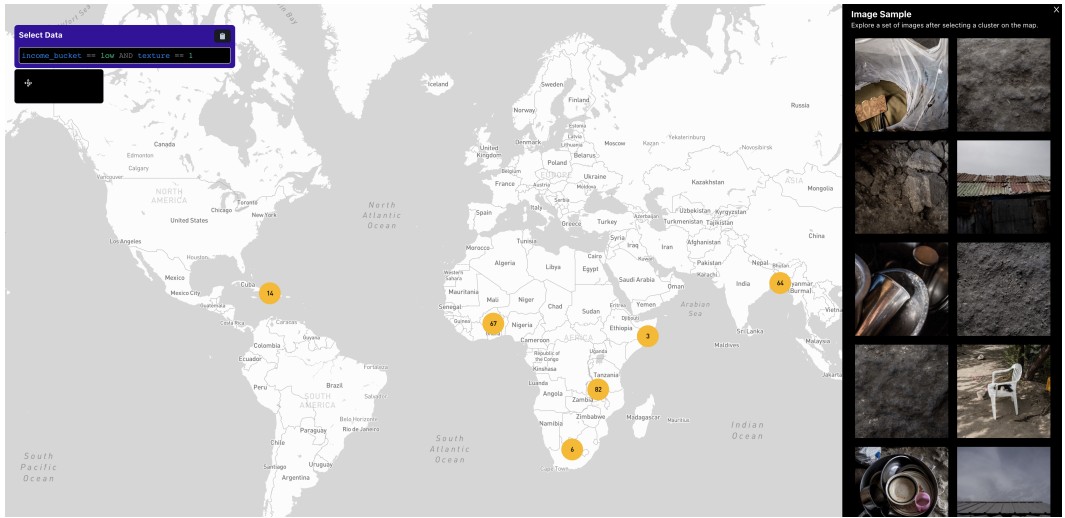

Figure 13: Interactive dashboard for Dollar Street factor annotations illustrating an example of the annotations.

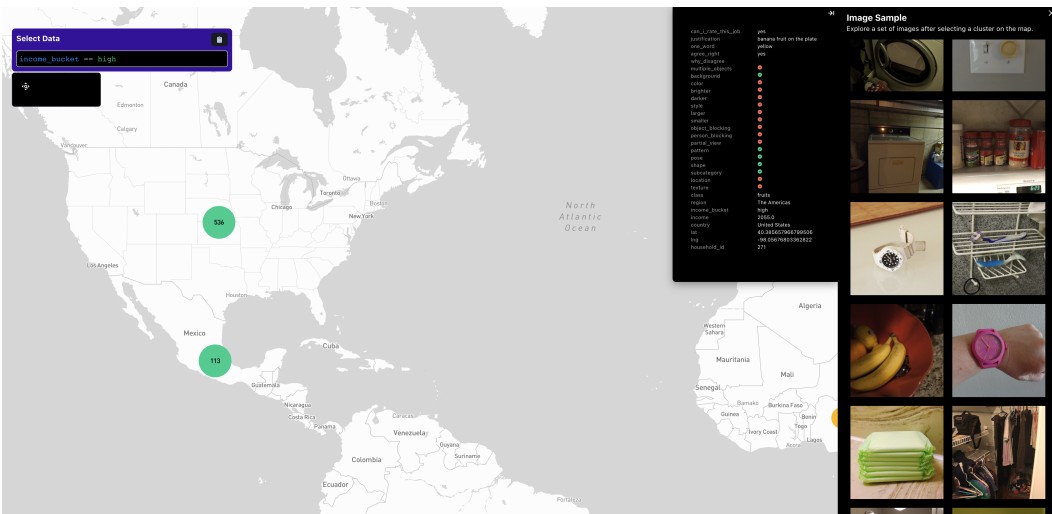

## B.2 Sample images

| Country | | | | |
|---|---|---|---|---|
| The Americas | | | | |
| Africa | | | | |

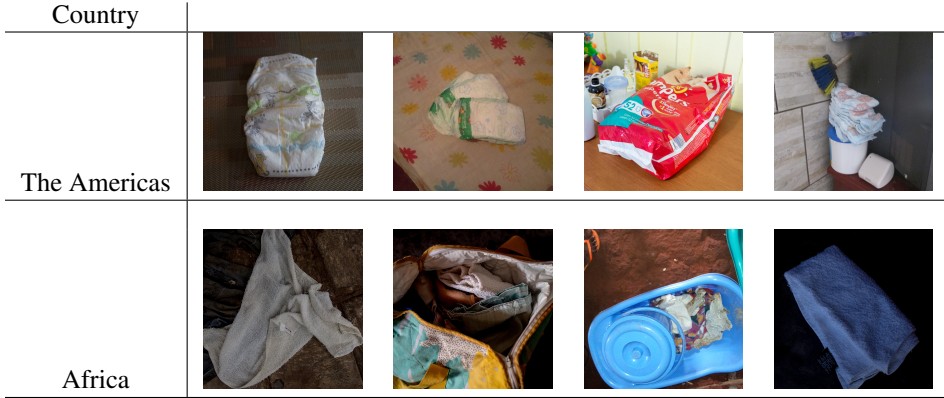

Table 17: Examples of diaper images. Our factors surfaced that images of diapers in Dollar Street between regions differed most among *pose*, *color*, *partial view*.

| Country | | | | |
|---|---|---|---|---|
| Asia | | | | |
| Africa | | | | |

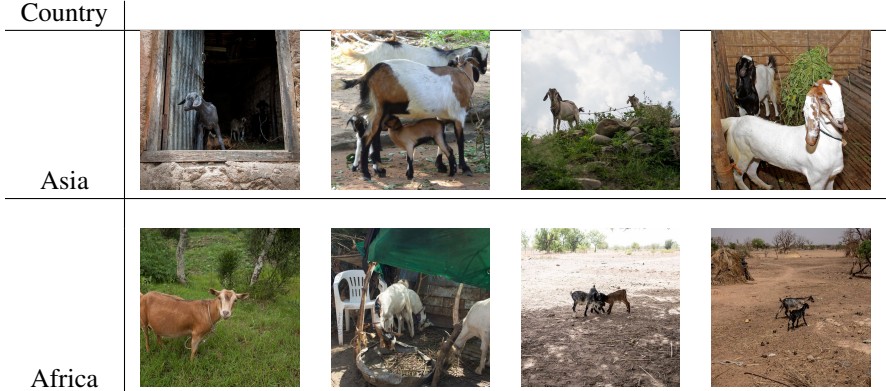

Table 18: Examples of goat images. Our factors surfaced that images of goats in Dollar Street between regions differed most among *pattern*, *color*, *subcategory*

In Tables 18 and 17 we show example images from classes and regions that were found to have some the starkest difference in factors, as measured by JSD.

