# OpenReview forum: "Exploring Why Object Recognition Performance Degrades Across Income Levels and Geographies with Factor Annotations"
_NeurIPS.cc/2023/Track/Datasets_and_Benchmarks — NeurIPS 2023 Datasets and Benchmarks Spotlight_

### Official Review · Reviewer_3kzU · 2023-07-21
**A benchamrk with a different point of view**

**Rating:** 9
**Confidence:** 5
**Correctness:** The reported findings are convincing.

**Strengths:**

1.	Providing annotations for images from Dollar Street with colors, shapes, poses, and backgrounds.
2.	Investigating performance disparities in models.
3.	Indicating ways for performance improvement related to texture bias.

**Additional Feedback:**

A solid work sharing new insight.

**Clarity:**

Related works are placed close to the Conclusions section. Please explain this step. Did you intend to achieve something by contrasting its content with the conclusions?

**Documentation:**

I was able to access the https://github.com/facebookresearch/dollarstreet_factors; however, the link https://dollarstreetfactors.dev.metademolab.com/ leads to an empty page (checked with different operating systems and browsers). The link in the supplement is valid https://dollarstreetfactors.metademolab.com.

**Ethics:**

44 annotators from 5 different countries participate in the labeling process.  Since the experiments involve human participation, it is important to share information on the agreement for such tests from the appropriate ethics committee. This part should be clarified to ensure transparency and compliance with ethical standards.

**Limitations:**

It is not clear why authors have focused on colors, shapes, poses, or backgrounds for the annotation. A justification is needed.

**Opportunities For Improvement:**

The study is well-thought-out and seems complete. However, the application of models that mitigate factors other than the texture bias would further support the findings. Is it possible?

**Relation To Prior Work:**

The limitations of previous studies are discussed.

**Summary And Contributions:**

In the paper, a new insight into how objects differ across incomes and regions is provided with the help of the annotated Dollar Street dataset and a set of supporting experiments.

---

> ### Author Response · Authors · 2023-08-23
>
> We’re very glad after a careful review you found our study to be well-thought out, complete, convincing, and offering new insights.
>
> > It is not clear why authors have focused on colors, shapes, poses, or backgrounds for the annotation.
>
> This is a great question. We acknowledge we should improve the clarity of how this choice is made. We now include several new sentences in our revised manuscript to ensure the motivation for this choice is appropriately emphasized. In our work, we study the full set of 16 factors based on the work of Idrissi et al. https://arxiv.org/abs/2211.01866. This set of factors was verified against free-form text responses collected from annotators based on ImageNet to ensure the 16 factors covered the most common distinctive image factors (see Section 2 of https://arxiv.org/abs/2211.01866). In our work, we additionally control for image quality (resolution in Section 4.1 and now also blurring)  to ensure image quality is not a confounding factor in our analysis. Please us know if the revisions and explanation above makes clear how the set of factors was selected to ensure we cover the most important image factors.
>
>
> > The application of models that mitigate factors other than the texture bias would further support the findings. Is it possible?
>
> We absolutely agree mitigating other factors would be an important next step towards the broader goal of improving foundation models’ performance disparities. While we did find a tailored mitigation for texture from https://arxiv.org/abs/1811.12231, to the best of our knowledge mitigations for generic factors remains an open research problem. We welcome suggestions and hope by releasing the entire set of factor annotations and our analysis to offer the research community a resource with which to explore other mitigations.
>
>
> > Related works are placed close to the Conclusions section. Please explain this step. Did you intend to achieve something by contrasting its content with the conclusions?
>
> Our aim is to contextualize the broader discussion of conclusions within the context of the existing literature. We’re open to incorporating reviewers’ suggestions if related works would be better suited to an earlier section of the work. Note we also now include a new discussion of limitations in Section 8.1.
>
> > Correct link to the factor annotations and demo
>
> Thank you for catching this mistake. We’ve now corrected the link everywhere to https://github.com/facebookresearch/dollarstreet_factors (and confirmed this works across different web browsers).
>
> > 44 annotators from 5 different countries participate in the labeling process. Since the experiments involve human participation, it is important to share information on the agreement for such tests from the appropriate ethics committee. This part should be clarified to ensure transparency and compliance with ethical standards.
>
> We thank you for pointing this out. We now include additional details about the annotators in Section 2.3 where we describe the country of origin is from Indonesia. We found that these annotators had a number of times that they disagreed with the class label. To control for regional differences in the perception of different object classes, we collected annotations about class-agreement (does the annotator agree with the DollarStreet class label) from a different set of 44 annotators from 5 different countries. We found that raters who were from the same geographical region as the image agreed with the class label approximately as frequently as those from other geographic regions (an average of 49.1% and 46.8% respectively). We now also include further discussion of some of the limitations associated with these regional comparison in our new Section 8.1.
>
>
> We thank you for the insightful feedback, recognition of the importance and quality of our work, and remain available for further discussions.

---

> > ### Comment · Reviewer_3kzU · 2023-08-29
> >
> > Thank you for your answers and modifications. In my opinion, the paper is worth reading.

---

### Official Review · Reviewer_uwZ7 · 2023-07-27
**An expanded Dollar Street dataset and analysis with factor annotation**

**Rating:** 7
**Confidence:** 3

**Strengths:**

This paper studies an interesting problem of what factors cause performance disparity across demographics e.g. income, region. The annotations are valuable information to help the community understand the failure modes of the models, and help develop methods that are more robust to these factors. The authors provide thorough analyses on these factors across various model architectures and pretraining objectives. This work has clear positive ethical/social implications to make foundational models more fair/equitable.

**Additional Feedback:**

Optional: the paper can use a teaser figure to highlight the key contributions of this paper (maybe a mix of Figure 1 and 2).

**Clarity:**

The paper writing is okay. Minor suggestion: I think the analyses section 3-6 can use some re-organization to group related subsections together to facilitate reading.

**Correctness:**

Yes, the claims are mostly correct. See Opportunities For Improvement section for more details.

**Documentation:**

Yes there are sufficient details on these. Licensing is in the supp. materials. Hosting/maintenance should be fine since this is just releasing the factor labels and visualization tool.

**Ethics:**

I think it should be fine.

**Limitations:**

The authors briefly mention the limitations in Conclusion of the manuscript. I think it'd be helpful if the authors can include separate sections on limitations and societal impact in the supp. materials or manuscript if space allows.

**Opportunities For Improvement:**

* Table 2 shows promising result with texture debiasing. How about adding more factors in figure 2 e.g. pose, subcategory, background? It would be interesting to use a more general technique to handle a broader range of factors presented in the paper. Another idea is to handle texture + darker lighting + occlusion which are the main factors for performance disparity.
* Many bars in Figure 2, 4, 6 appear close to each other. It'd be helpful to see error bars so that readers know if the differences are meaningful or not. For example, how does the disagreement rate in Table 6 (supp. materials) affect the bars we're seeing?
* The claims that factor labels can explain model performance disparity may not be entirely true because correlation is not the same as causality, unless there's other evidence proving so. For example, if you look at the distribution of feature vectors within a class instead of factor label distribution, would you derive the same conclusion as Figure 5?
* In Figure 2, pose, background, pattern, and color all seem very similar across demographics. The more interesting ones that differ across demographics (e.g. texture) are very small and barely visible. Maybe there is a better way to visualize these factors.
* Figure 4 misses X-axis.

**Relation To Prior Work:**

Yes it is.

**Summary And Contributions:**

This work annotates the Dollar Street dataset with many factors (i.e. what sets this image apart from the prototype of this class) to analyze the vulnerabilities of existing pretrained models (e.g. CLIP) across different fairness axes.  The contribution is the annotation itself, analysis on failure modes/factors, finding that mitigation of texture can improve robustness, and visualization dashboard.

---

> ### Author Response · Authors · 2023-08-23
>
> We thank you for carefully reviewing our work and for offering several suggestions to further improve the clarity and scope of our work. We’re glad to see you found our work to contribute a valuable insights to the research community including the factor annotations, analysis of texture bias mitigation, and visualization. We’re thrilled you find this work to have a clear positive ethical and social impact to make foundation models more equitable and fair. We’ve incorporate several of your suggestions and address specific comments below.
>
> > It would be interesting to use a more general technique to handle a broader range of factors presented in the paper.
>
> We absolutely agree exploring a more general technique to handle a broader range of factors would be an interesting followup. While we found a tailored mitigation for texture from https://arxiv.org/abs/1811.12231, in our investigation of other mitigation techniques we did not find general purpose solutions to mitigate broader vulnerabilities in object recognition systems. To the best of our knowledge such general purpose mitigations remain an open research problem. We’d be glad to implement other mitigation methods to tackle a broader range of factors if the reviewer has any specific suggestions.
>
> While we show promise via texture debiasing, we wholeheartedly agree a more general solution would be important step towards ensuring object recognition systems perform well across geographies and incomes. To this end, our goal in releasing the entire set of factor annotations and analysis is to do just that: we surface for the first time factors associated with such  performance degradations precisely in order to spur directions for future research into general solutions.
>
> > Many bars in Figure 2, 4, 6 appear close to each other. It'd be helpful to see error bars so that readers know if the differences are meaningful or not.
> > In Figure 2, pose, background, pattern, and color all seem very similar across demographics.
> > For example, how does the disagreement rate in Table 6 (supp. materials) affect the bars we're seeing?
> Thank you for pointing this out. We agree this presentation is potentially confusing. Unless otherwise noted, the marginal differences between bars in Figures 2, 4, 6 are not meaningful. Instead, the figures are intended to illustrate the relative rank of each factor. For example, in Figure 4 we wish to illustrate that across the range of model architectures and learning procedures, the drop in accuracy across regions and incomes is quite similar. We now provide more context to clarify the differences for annotator disagreement rates (show in Appendix Table 6) in Section 2 and 8.1 highlighted in blue.
> Where were wish to isolate meaningful differences is in model vulnerabilities (Figure 6 and Figure 7). In these cases, we plot the error ratio which describes how much more likely a factor is to appear among a model’s mistakes. We quantitatively describe such differences: “texture is 1.7x more likely to appear among a models’ mistakes” and substantiate the claim that such differences are meaningful with statistical testing. For example, we conduct a Chi-squared test to verify differences in factor error ratios are statistically significant (see Appendix A.4).

---

> > ### Author Response · Authors · 2023-08-23
> >
> > > The claims that factor labels can explain model performance disparity may not be entirely true because correlation is not the same as causality, unless there's other evidence proving so.
> >
> > This is a keen point. We agree correlation does not imply causation. An inherent limitation of both our analysis and the broader subfield of human-friendly interpretability methods for deep learning models (https://proceedings.mlr.press/v80/kim18d/kim18d.pdf). To our knowledge this is the first attempt to surface insights into vulnerabilities underlying performance disparities across incomes and geographies. We surface these vulnerabilities based on human annotated factors using the error ratio https://arxiv.org/abs/2211.01866. The error ratio identifies how much more likely a factor is to appear among a models mistakes relative to overall among a model’s predictions. While such analysis does not guarantee causal factors, we verify vulnerabilities are statistically significant by comparing likelihoods within misclassifications relative to the overall set of predictions (Appendix A.4). Finally, our promising mitigation for texture suggests we can intervene on the underlying factors to improve performance disparities (+4.1% boost in accuracy for low income). Based on your feedback, we now explicitly acknowledge this limitation explicitly in Section 4.3:
> >
> > Added the following sentences in Section 4.2: “We acknowledge such an analysis of likelihoods is inherently not causal.” and that are “based on associations between factors and mistakes.”
> > We hope this more appropriately aligns the limitation of our analysis with the presentation of the error ratio analysis in our work. Please let us know if have have further suggestions.
> >
> > > Figure 4 misses X-axis.
> >
> > We appreciate you pointing out this mistake. We’ve updated the x-axis for Figure 4 in revised manuscript on OpenReview based on this feedback.
> >
> > > Minor suggestion: I think the analyses section 3-6 can use some re-organization to group related subsections together to facilitate reading.
> >
> > Thanks for the suggestion to improve the flow of the paper. Based on this suggestion, we’ve reordered the flow of the paper between Sections 3 and 4 for improved clarity. We now first discuss modern models’ performance across incomes and geographies to then motivate our study of how factors differ across incomes and geographies. We hope this improves the flow and clarity of our work and welcome any further suggestions.

---

> > > ### Comment · Reviewer_uwZ7 · 2023-08-29
> > > **Thank you for the feedback**
> > >
> > > The authors' feedback has addressed my concerns. I've raised the score to 7: "Accept". Thank you.

---

### Official Review · Reviewer_cjSi · 2023-07-31
**Review of the paper**

**Rating:** 7
**Confidence:** 4
**Clarity:** The paper is clearly written.

**Strengths:**

- The paper studies an important yet less studied problem in computer vision and deep learning.
- The annotated data could be quite useful for the community.
- The analysis and initial experiments in mitigating the issue with texture debiasing provide insights for the community.

**Additional Feedback:**

No additional feedback.

**Correctness:**

The reviewer believes the dataset construction, the benchmarking, and the analysis are technically sound.

**Documentation:**

The reviewer believes sufficient details are provided. The webpage is well-made and provides an intuitive way to explore the dataset.

**Ethics:**

The reviewer do not think this paper has major ethics concerns.

**Limitations:**

The reviewer does not think there is a significant limitation in the current scope of the paper. In the future, if we want to expand the scope of the study, there could be some limitations we need to consider:
1. Whether the current annotations collected are sufficient for other related computer vision problems, such as object detection, and image segmentation; or other applications such as autonomous driving? Do we need to collect other factors?
2. Texture debiasing is interesting but the help it can provide seems marginal. What could be other more effective ways to help mitigate the issue?

**Opportunities For Improvement:**

- Adding more factors. For example, image resolution is studied in Section 5.1, but how about other image quality factors such as blur and out of focus, etc.?
- Minor: There are no labels for the horizontal axis in Figure 4.

**Relation To Prior Work:**

The reviewer believes the related work is sufficiently discussed.

**Summary And Contributions:**

This paper studies the problem of why modern image classification models have large performance discrepancies across income levels and geographic regions. The authors annotate DollarStreet images with factors that can potentially contribute to such performance discrepancies. Equipped with rich annotations, the authors performed a list of experiments using off-the-shelf deep learning based image classification models such as CLIP and ViT. Some interesting observations include: 1) texture, occlusion, and darker lighting are most associated with models’ performance degradation; 2) model architecture and training are not essential factors.

The authors also found that performance drop is correlated with factor dissimilarity percentage, showing the usefulness of collected annotations in indicating the model's performance. Furthermore, the authors showed that texture debiasing can be used to improve the model's performance overall, especially for the low-income group.

All the annotations are released and aggregated into a webpage (https://dollarstreetfactors.metademolab.com/). The reviewer believes the paper studied an important problem. The collected data and the analysis provided insights into the community.

---

> ### Author Response · Authors · 2023-08-23
>
> Thank you for such a careful review of our work. We very much appreciate that you find we study an important yet understudied problem in computer vision, that our paper is clearly written, and that the factor annotations we release along with the website are well-made, intuitive and valuable to the research community.
>
> > What about studying other image quality factors such as blurr or out-of-focus?
>
> This is a great point: studying additional image quality factors sucha s blurr would improve the comprehensiveness of the factors we study. In our work, we study the full set of 16 factors based on the work of Idrissi et al. https://arxiv.org/abs/2211.01866. This set of factors was verified against free-form text responses collected from annotators based on ImageNet to ensure the 16 factors covered the most common distinctive image factors (see Section 2 of https://arxiv.org/abs/2211.01866). In our work, we additionally control for image resolution in Section 5.1 to ensure image quality is not a confounding factor in our analysis. We agree blurring and focus are two additional important factors related to image quality.
>
> | blurring range     | number of samples |
> | ------------------ | ----------------- |
> | (0.80, 3536.9)     | 13512             |
> | (3536.9, 7073.0)   | 421               |
> | (7073.0, 10609.1)  | 53                |
> | (10609.1, 14145.2) | 6                 |
> | (14145.2, 17681.3) | 4                 |
>
> Table 1: The bins and number of samples in a 5 binned histogram for variation of the Laplacian across images.
>
> Based on your feedback we ran an additional analysis based on automatic detection of image blurring. Following https://www.sciencedirect.com/science/article/pii/S0031320312004736,  we measure the blurriness of an image as the variation of the Laplacian for the gray-scaled image. We find this blurr measure has extremely high variation across the DollarStreet dataset as shown in Table 1 above (with a standard deviation of 1124 that’s larger the mean). Qualitative inspection also revealed blurry images often had a blurry background but a sharp focus for the primary object, suggesting overall blurriness does not necessarily correspond to a blurry object. We found a similarly high standard deviation when grouping by income or region, suggesting we can not draw conclusions about meaningful differences in blurriness across incomes or regions.
>
> We hope this additional analysis of blurring/focus as well as our clarification of the choice of factors used in our annotations address your comment. We remain available for further discussion or to answer any remaining questions.
>
> > Figure 4 axis is missing.
>
> Thank you for catching this. We now include an updated Figure 4 in the revised manuscript.
>
> > Expand scope of study: could we use the current annotations for other computer vision tasks such as object detection or application such as autonomous driving? Do we need to collect other factors?
>
> We absolutely agree expanding the discussion of how these factors could be applied to other tasks would greatly expand the scope of this work. Thank you for bringing this to our attention.  While the factors are based class groupings (using the ground-truth class labels from DollarStreet), the questions we ask annotators identify what is most distinctive about each image. Therefore, we do believe the factors could be useful for applications beyond standard classification such anomaly detection or interpretability and potentially for coarse grained signals of failure modes for object detection. However, the utility of these factors would be somewhat limited depending on how specialized the task at hand is and the stakes involved. In particular for high-stakes applications such as autonomous driving, while these factors may provide useful coarse grained signals, we’d hypothesize richer bounding boxes based annotations with domain-specific annotation questions weighing the appropriate precision-recall tradeoff risks would be necessary. Based on your feedback, we now include a lengthier discussion of limitations and the scope of applications where our factors would be best suited in a new limitations section (Section 8.1, highlighted in blue).

---

> > ### Author Response · Authors · 2023-08-23
> >
> > > Expand scope of study: Texture debiasing is interesting but the help it can provide seems marginal. What could be other more effective ways to help mitigate the issue?
> >
> > We absolutely agree exploring a more general technique to handle a broader range of factors would be an interesting followup. Based on your feedback we did uncover one other possible solution studied in https://arxiv.org/abs/2307.13136  which retrains the last layer on balanced data (by region or income). This approach requires gathering additional training data and unfortunately does not target the underlying factor vulnerabilities. While we found a tailored mitigation for texture from https://arxiv.org/abs/1811.12231, in our investigation of other mitigation techniques we did not find general purpose solutions to mitigate broader factor vulnerabilities in object recognition systems. To the best of our knowledge such general purpose mitigations remain an open research problem. Given the importance this problem, we hope by releasing the entire set of factor annotations and our analysis to offer the research community a resource with which to explore other mitigations.
> >
> > We remain available to clarify any remaining questions and for further discussion.

---

> > > ### Comment · Reviewer_cjSi · 2023-08-23
> > >
> > > Thanks for the authors' response. I believe my questions were well addressed. The newly added Sec. 8.1 looks great!

---

### Author Response · Authors · 2023-08-23

We thank reviewers for carefully assessing our work and providing several excellent suggestions. We’re thrilled all reviewers recognized we investigate an “important yet understudied problem” (Reviewer cjSi) and provide “quite useful insights for the community” via “technically sound” (Reviewer cjSi), “complete”, and  “convincing” (Reviewer 3kzU) analysis with “clear positive ethical/social implications to make foundation models more fair” (Reviewer uwZ7). We’re also glad reviewers found our paper to be “clearly written” (Reviewer cjSi) and the webpage to be “well-made” and “intuitive” (Reviewer cjSi).

Thanks to reviewers’ feedback we’ve now clarified our choice of factors (thanks to Reviewers 3kzU and cjSi), produced a new analysis of image quality blurring/focus factors (thanks to Reviewer cjSi), made several additions to the manuscript (highlightedd in blue) including an expanded discussion of limitations throughout the text and in a new conclusion section 8.1 (thanks to Reviewers cjSi and uwZ7), and reorganized Sections 3 and 4 for improved flow (thanks to Reviewer uwZ7) among other changes. These revisions thanks to reviewers’ feedback have improved both the quality and clarity of our work. We very much appreciate the thoughtful feedback from all reviewers.

We remain available for further discussion and to address any outstanding comments from reviewers.

---

### Decision · Program_Chairs · 2023-09-22

**Decision:**

Accept (Spotlight)

**Comment:**

The paper has garnered unanimous recommendations for acceptance, including two acceptances and one strong acceptance. The paper's focus on evaluating the performance of machine learning systems across different geographies and income levels. The research addresses an important yet under-explored area of research. Reviewers concur that the analysis and data collected offer valuable new insights for the research community and carry significant ethical and social implications, particularly in the pursuit of making foundational models more fair and equitable. The Area Chair (AC) aligns with the reviewers' assessments and supports their recommendation to accept the paper.